# REASONING VECTORS: TRANSFERRING CHAIN-OF-THOUGHT CAPABILITIES VIA TASK ARITHMETIC

## ABSTRACT

Large language models often require costly optimization, such as reinforcement learning, to master complex reasoning tasks. This work demonstrates that reasoning ability, once learned, can be extracted and transferred between models as a compact task vector. We source two publicly available, identically initialized QWEN2.5 models, one fine-tuned with supervised fine-tuning (SFT) and the other with group relative policy optimization (GRPO) on the same dataset. From these, we extract a *reasoning vector*: $\mathbf{v}_{\mathrm{reason}} = \theta_{\mathrm{GRPO}} - \theta_{\mathrm{SFT}}$. We hypothesize that this vector captures the reasoning capability instilled by reinforcement learning while factoring out shared knowledge from the SFT process. When added to compatible instruction-tuned models through simple arithmetic, this vector consistently improves performance across diverse reasoning benchmarks: GSM8K (+4.9%), HumanEval (+4.3%), SciQ (+1.7%), and BigBenchHard (+12.3% for the 1.5B model). The performance improvements persist under adversarial conditions. Conversely, subtracting the vector causes significant performance degradation (-11.8% on GSM8K), demonstrating the vector's strong contribution to the model's reasoning abilities. This work shows how reasoning capabilities, typically developed through expensive training, can be extracted from existing open-source models and reused through simple tensor arithmetic, offering a practical way to enhance models by recycling prior computational investments.

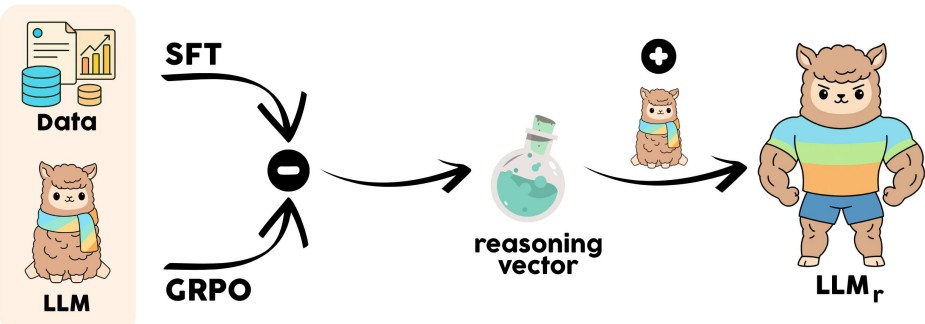

Figure 1: **Merging the Fine-Tuning and Reasoning Vectors.** Let $\Delta_f = \theta_f - \theta_0$ denote the fine-tuning vector ($f$) and $\mathbf{v}_{\mathrm{reason}} = \theta_r - \theta_f$ denote the reasoning vector ($r$). By adding $\mathbf{v}_{\mathrm{reason}}$ to a base model, we obtain an enhanced model with improved reasoning capabilities, effectively transferring the outcome of the reinforcement learning phase.

## 1 INTRODUCTION

Large language models (LLMs) excel at knowledge retrieval, but often falter on multistep reasoning tasks. While training-time methods like reinforcement learning from human feedback (RLHF)

[23; 36] robustly improve reasoning, they demand substantial computational resources and intricate tuning. This high cost limits their broad application, creating the need for more accessible methods to enhance reasoning.

Task arithmetic [13] presents a paradigm in which the capabilities learned during fine-tuning can be represented as vectors and transferred between models by arithmetic. This raises a compelling question: *Can complex reasoning capabilities, acquired through resource-intensive reinforcement learning, be extracted from existing models and transferred as reusable task vectors?*

This paper answers affirmatively by introducing the concept of a *reasoning vector*. Our approach takes advantage of the growing availability of open-source models. We source two models with identical initialization and training data history, differing only in their final optimization stage: one with supervised fine-tuning ($\theta_{\text{SFT}}$) and the other with Group Relative Policy Optimization ($\theta_{\text{GRPO}}$). We define the reasoning vector as their difference: $\mathbf{v}_{\text{reason}} = \theta_{\text{GRPO}} - \theta_{\text{SFT}}$. This controlled subtraction aims to isolate the parameter changes associated with the RL-induced reasoning enhancements, while minimizing the influence of dataset-specific knowledge shared by both models. Adding this vector to a compatible instruction-tuned model allows for the transfer of reasoning abilities without requiring any new training. This is illustrated in Figure 1.

This method allows for the reuse of the significant computational effort already invested in training advanced models. By sourcing checkpoints from public model hubs, one can enhance a base model's reasoning capabilities through a few simple tensor operations. Our experiments on QWEN2.5 models (1.5B and 7B parameters) show consistent performance improvements across diverse reasoning benchmarks. For the 1.5B model, adding the reasoning vector improves accuracy on GSM8K by 4.9%, HumanEval by 4.3%, and BigBenchHard by 12.3%. These gains hold under adversarial perturbations. An ablation study confirms the vector's impact: its removal degrades performance on GSM8K by 11.8%, falling below the SFT baseline.

Our contributions are:

1. We demonstrate that a reasoning capability associated with reinforcement learning can be extracted as a modular vector component from existing, publicly available models.

2. We show that a reasoning vector derived from mathematical training data generalizes to improve performance on other domains, including code generation, scientific QA, and logical deduction.

3. We provide a reproducible method that leverages open-source checkpoints and simple tensor arithmetic, increasing the accessibility of reasoning-enhanced models by promoting the reuse of existing resources.

## 2 RELATED WORK

**Reasoning in Large Language Models.** Enhancing reasoning in LLMs follows two primary approaches. *Prompting strategies* generate reasoning from existing parameters: chain-of-thought prompting [29] encourages step-by-step verbalization, self-consistency [28] samples multiple reasoning paths, tree-of-thoughts [33] explores branching logic, and zero-shot reasoning triggers like "think step by step" [15] activate latent capabilities. Program-aided approaches, such as Program-of-Thought (PoT) [3] and Program-Aided Language Models (PAL) [8], offload computation to external interpreters. *Training-based methods* directly encode reasoning through supervised fine-tuning on annotated datasets [5; 10] or reinforcement learning, including RLHF [23; 4], PPO [26], and GRPO [24]. Some methods combine RL with verifier models [5; 17] to evaluate reasoning steps. Despite progress, benchmarks reveal that even advanced models can struggle with complex multistep reasoning. Our work bridges these approaches: we extract the capabilities learned through RL and transfer them without requiring additional training cycles.

**Task Arithmetic and Model Merging.** Task arithmetic [13] demonstrates that fine-tuning capabilities can be represented as vectors in parameter space and composed through arithmetic operations. Extensions include TIES-Merging [31], which reduces interference by resolving sign conflicts; Fisher merging [19], which weights parameters by their importance; and RegMean [14], which formulates optimal parameter combinations as a regression problem. Model soups [32] average model weights for single-task improvement, while methods like Ratatouille [25] target out-of-domain generalization.

Linear Mode Connectivity [7; 1] provides a theoretical foundation, showing that models fine-tuned from an identical initialization lie in connected low-loss regions, enabling safe linear interpolation. Recent work has scaled these techniques to billion-parameter models [12; 34], with practical tools like `MergeKit` [9] operationalizing parameter space composition. While prior work focuses on domain knowledge or task-specific skills, we investigate whether a complex cognitive capability like multi-step reasoning, acquired via RL, can similarly be isolated and transferred.

**Modular Capability Enhancement.** Parameter-efficient methods add small, trainable components to frozen models: LoRA [11] introduces low-rank adaptation matrices, prefix tuning [18] prepends learnable tokens, and prompt tuning [16] optimizes continuous prompts. Knowledge editing techniques [20] modify specific facts by targeting individual neurons, while tangent space methods [22] aim to improve weight disentanglement for better merging. Instruction tuning has become particularly amenable to merging, with studies showing that averaging instruction-tuned experts can outperform standard multitask training [37; 2]. Unlike these approaches, which often require task-specific training or narrow edits, our method explores a training-free, global enhancement. We examine if a reasoning vector derived from mathematical training (GSM8K) can generalize to improve code generation, scientific QA, and logical deduction, suggesting that reasoning can be treated as a transferable capability distinct from task-specific patterns [35; 30].

## 3 METHODOLOGY

### 3.1 PROBLEM FORMULATION

Consider two models sourced from a public repository that share an identical architecture, initialization, and pre-training history. Let $\theta_{\text{SFT}}$ denote the parameters of a model that has undergone supervised fine-tuning on a specific dataset $\mathcal{D}$ (e.g., the GSM8K training set) using a standard cross-entropy loss. Let $\theta_{\text{GRPO}}$ denote the parameters of a counterpart model that was optimized using Group Relative Policy Optimization (GRPO), a reinforcement learning algorithm, on the same dataset $\mathcal{D}$ with a reasoning-focused reward function. This controlled comparison allows us to isolate the impact of the reasoning vector itself indepedent of the pretraining dataset.

### 3.2 REASONING VECTOR EXTRACTION AND TRANSFER

We define the *reasoning vector*, $\mathbf{v}_{\text{reason}}$, as the difference in parameters between these two models:

$$\mathbf{v}_{\text{reason}} = \theta_{\text{GRPO}} - \theta_{\text{SFT}} \tag{1}$$

We hypothesize that this vector, $\mathbf{v}_{\text{reason}}$, captures the essential parameter updates introduced by the reinforcement learning process that enhance multi-step reasoning. Because both donor models share the same data and base knowledge, the subtraction is intended to factor out this shared, dataset-specific information, leaving behind a more general representation of the reasoning capability.

Given a target instruction-tuned model, $\theta_{\text{target}}$, that is compatible with the donor models, we can enhance its reasoning ability through a simple arithmetic operation:

$$\theta_{\text{enhanced}} = \theta_{\text{target}} + \alpha \cdot \mathbf{v}_{\text{reason}} \tag{2}$$

where $\alpha \in [0, 1]$ is a scalar coefficient that controls the magnitude of the transferred vector. For more fine-grained control, this operation can be applied to specific layers or modules by introducing a binary mask $\mathbf{m} \in \{0, 1\}^{|\theta|}$:

$$\theta_{\text{enhanced}} = \theta_{\text{target}} + \alpha \cdot (\mathbf{m} \odot \mathbf{v}_{\text{reason}}) \tag{3}$$

where $\odot$ denotes element-wise multiplication. In our experiments, we found that applying the full vector ($\mathbf{m} = \mathbf{1}$) with a scaling factor of $\alpha = 1$ was consistently effective, suggesting the extracted vector is well-calibrated for transfer without needing further adjustment.

### 3.3 THEORETICAL FOUNDATION

The safety and effectiveness of this transfer relies on the principle of Linear Mode Connectivity (LMC) [7]. LMC states that when two models are fine-tuned from the same initialization, they

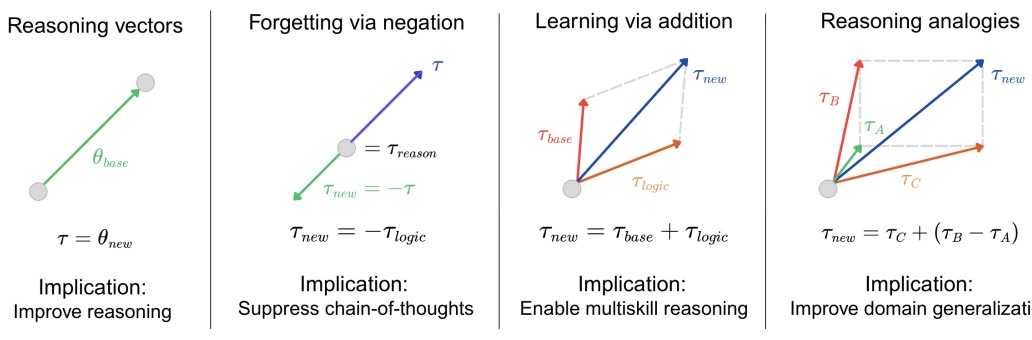

Figure 2: **Reasoning vector operations in weight space.** Each panel illustrates a different transformation: (1) *Vector injection* shifts a base model toward improved reasoning. (2) *Negation* removes the reasoning component, suppressing chain-of-thought behavior. (3) *Addition* combines multiple skill vectors, enabling multi-skill reasoning. (4) *Analogy-style composition* transfers capabilities across domains, supporting generalization.

typically lie in the same connected low-loss basin of the optimization landscape. Formally, for parameters $\theta_A$ and $\theta_B$ obtained from the same starting point, their convex interpolation satisfies:

$$\mathcal{L}(\lambda\theta_A + (1-\lambda)\theta_B) \leq \max\big(\mathcal{L}(\theta_A), \mathcal{L}(\theta_B)\big) + \epsilon, \quad \lambda \in [0, 1], \tag{4}$$

where $\mathcal{L}$ is the loss function and $\epsilon$ is a small value.

Intuitively, this inequality guarantees that the straight line between $\theta_A$ and $\theta_B$ in weight space does not leave the low-loss region; moving continuously from one model to the other does not increase the loss beyond that of the worse endpoint. In geometric terms, both models occupy the same flat "valley" of the loss surface, and the connecting path avoids high-loss barriers.

Because $\theta_{\mathrm{SFT}}$ and $\theta_{\mathrm{GRPO}}$ share the same initialization and were trained on the same data, they are expected to satisfy the conditions for LMC. Their difference vector,

$$\mathbf{v}_{\mathrm{reason}} = \theta_{\mathrm{GRPO}} - \theta_{\mathrm{SFT}}$$

therefore points in a direction within this shared low-loss basin. Adding this vector to another compatible model corresponds to moving it along a trajectory that has been implicitly validated to remain within a stable, low-loss region. This explains why the transfer is effective and can enhance reasoning ability without catastrophically destabilizing the base model's existing capabilities.

### 3.4 IMPLEMENTATION DETAILS

The complete procedure for enhancing a model requires only two tensor operations, as summarized below.

$$\text{Extract:} \quad \mathbf{v}_{\mathrm{reason}} \leftarrow \theta_{\mathrm{GRPO}} - \theta_{\mathrm{SFT}} \tag{5}$$
$$\text{Transfer:} \quad \theta_{\mathrm{enhanced}} \leftarrow \theta_{\mathrm{target}} + \mathbf{v}_{\mathrm{reason}} \tag{6}$$

These operations are element-wise and computationally inexpensive. The vector $\mathbf{v}_{\mathrm{reason}}$ can be pre-computed and stored, then applied to any number of compatible target models on demand.

**Compatibility Requirements.** For a successful transfer, the target model must satisfy:

1. **Architecture Match:** Identical layer structures, hidden dimensions, and parameter tensor shapes.

2. **Tokenizer Compatibility:** The same vocabulary and token-to-ID mapping to ensure semantic alignment, especially in the embedding layer.

3. **Initialization Similarity:** The models should ideally originate from the same pre-trained checkpoint family to ensure their parameter spaces are sufficiently aligned.

**Activation Strategy.** While the enhanced models demonstrate improved performance intrinsically, we find that prefixing input prompts with a simple instruction like "Think step by step" reliably activates and enhances the transferred reasoning capability. This minimal cue appears to prime the model to leverage the problem-solving pathways encoded in the reasoning vector, analogous to how RL-trained models are conditioned to engage in systematic reasoning.

# 4 EXPERIMENTS

We evaluate the effectiveness of reasoning vector transfer across multiple dimensions: performance on core reasoning benchmarks, robustness under adversarial conditions, and ablations to understand the vector's impact. Our experiments utilize QWEN2.5 models at 1.5B and 7B parameter scales.

## 4.1 EXPERIMENTAL SETUP

**Model Configuration.** We use publicly available QWEN2.5 models at 1.5B and 7B scales. For each size, our donor models consist of a checkpoint fine-tuned on the GSM8K training split via SFT ($\theta_{\text{SFT}}$) and a counterpart further trained on the same data with GRPO ($\theta_{\text{GRPO}}$). The reasoning vector $\mathbf{v}_{\text{reason}} = \theta_{\text{GRPO}} - \theta_{\text{SFT}}$ is extracted and then added to the corresponding official QWEN2.5-Instruct base model ($\theta_{\text{target}}$) using the `MergeKit` library [9].

**Evaluation Configurations.** To isolate the effects of the vector and prompting, we compare four configurations across all benchmarks:

- **Baseline**: The original instruction-tuned QWEN2.5-Instruct model without modification.
- **G+T**: The GRPO-tuned donor model prompted with "Think step by step." This column serves as a reference for the performance of the RL-tuned source model.
- **+Vector**: The baseline model enhanced with the reasoning vector via addition ($\alpha = 1$).
- **+Vector+Think**: The vector-enhanced model evaluated with the prefix "Think step by step".

**Benchmarks.** We evaluate performance on five diverse, reasoning-oriented benchmarks:

- **GSM8K** [5]: Multi-step arithmetic reasoning on grade-school math problems.
- **HumanEval** & **HumanEval+** [6]: Python code generation from natural language docstrings, with HumanEval+ offering more rigorous tests.
- **SciQ** [21]: Multiple-choice science questions requiring domain knowledge and reasoning.
- **BigBenchHard** [27]: A suite of tasks designed to be difficult for LLMs, focusing on multi-hop reasoning, logic, and symbolic manipulation.

**Evaluation Protocol.** Accuracy is the primary metric, with pass@1 used for code generation tasks. For reproducibility, we use greedy decoding ($T = 0$) for deterministic tasks (GSM8K, SciQ, BigBenchHard) and set temperature to $T = 0.5$ for creative generation (HumanEval, HumanEval+). We use the standardized prompt templates shown in Figure 3 to ensure consistency. We report results from a single run for each experiment and acknowledge that multi-run evaluations would be needed to establish statistical significance.

## 4.2 MAIN RESULTS

Table 1 and Figure 4 present the performance of our method across all benchmarks. The results show a consistent positive trend: adding the reasoning vector generally improves model performance, with further gains often achieved by adding a simple reasoning prompt.

For the **1.5B model**, vector injection alone boosts GSM8K accuracy from 45.1% to 47.7% (+2.6%). When combined with a reasoning prompt, the accuracy reaches 50.0%, a total improvement of +4.9%. This positive trend holds across other domains. On HumanEval, the vector provides a +2.2% gain, increasing to +4.3% with prompting. Most strikingly, on the complex BigBenchHard

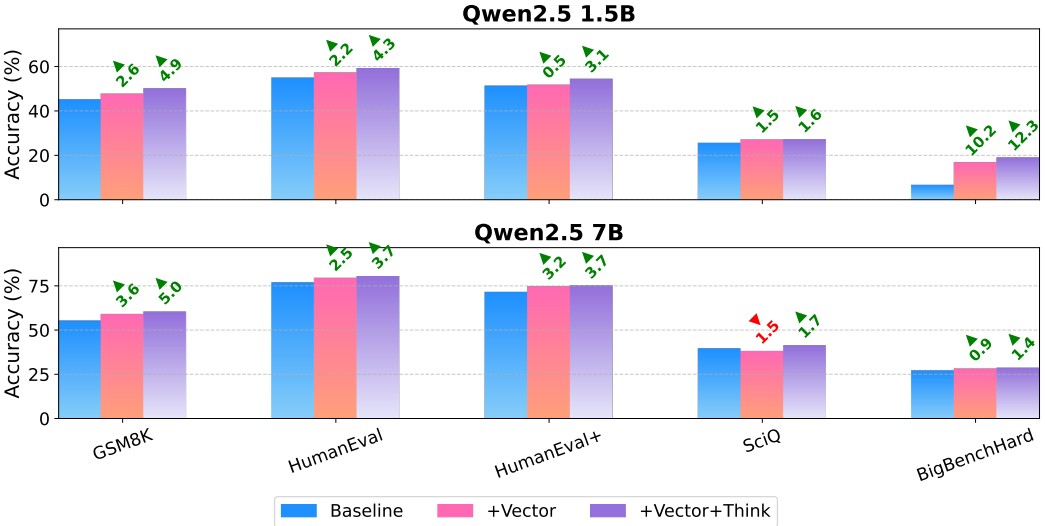

Figure 3: **Evaluation prompt templates for HumanEval and GSM8K.** This design allows us to distinguish between improvements from parameter modification alone versus those elicited by explicit reasoning prompts.

Figure 4: **Accuracy improvements from reasoning vector transfer.** Performance of QWEN2.5 models (1.5B left, 7B right) on five benchmarks. Bars compare baseline (blue), vector-enhanced (+Vector, green), and vector with prompt (+Vector+Think, orange). Green annotations show absolute accuracy gains. The vector transfer consistently improves performance, especially on complex benchmarks like BigBenchHard (BBH).

benchmark, the vector enhances the model's performance from a near-random 6.7% to 19.0% (+12.3%), demonstrating a substantial improvement in challenging reasoning scenarios.

The **7B model** exhibits similar improvements at a higher performance baseline. On GSM8K, the vector and prompt together increase accuracy from 55.3% to 60.3% (+5.0%). On HumanEval, performance reaches 80.5% (+3.7%). While the vector addition alone led to a minor performance drop on SciQ for the 7B model (-1.5%), this was recovered and surpassed when a reasoning prompt was included, resulting in a net gain of +1.7%. Overall, the addition of the reasoning vector provides a consistent and positive impact, indicating the method is robust across different model scales.

### 4.3 ROBUSTNESS ANALYSIS

To test whether the performance gains stem from genuine reasoning enhancement rather than superficial pattern matching, we evaluated the 1.5B model on three custom, adversarially modified versions of the GSM8K dataset.

**Perturbation Design.** We created three challenging variants:

Table 1: **Accuracy (%) of Qwen2.5 models on reasoning benchmarks.** Results show consistent improvements from reasoning vector injection. Green text indicates improvements over baseline, red indicates degradation. Absolute change is in parentheses. All results are from a single run.

| Benchmark | Qwen2.5 1.5B | | | | Qwen2.5 7B | | | |
|---|---|---|---|---|---|---|---|---|
| | Base | G+T | +Vec | +Vec+T | Base | G+T | +Vec | +Vec+T |
| GSM8K | 45.1 | 46.8 | 47.7 (+2.6) | **50.0** (+4.9) | 55.3 | 57.3 | 58.9 (+3.6) | **60.3** (+5.0) |
| HumanEval | 54.9 | 51.8 | 57.1 (+2.2) | **59.2** (+4.3) | 76.8 | 76.2 | 79.3 (+2.5) | **80.5** (+3.7) |
| HumanEval+ | 51.2 | 49.3 | 51.7 (+0.5) | **54.3** (+3.1) | 71.3 | 72.0 | 74.5 (+3.2) | **75.0** (+3.7) |
| SciQ | 25.6 | 25.8 | 27.1 (+1.5) | **27.2** (+1.6) | 39.6 | 37.0 | 38.1 (-1.5) | **41.3** (+1.7) |
| BigBenchHard | 6.7 | 16.7 | 16.9 (+10.2) | **19.0** (+12.3) | 27.3 | 28.2 | 28.2 (+0.9) | **28.7** (+1.4) |

- **GSM Hard Lite**: Problems with extended numerical ranges and more reasoning steps, increasing computational and logical complexity.

- **Noise+Digit**: The injection of irrelevant numerical tokens, typos, and distracting punctuation to test the model's focus.

- **Sentence Shuffle**: The order of sentences within a problem is altered while preserving logical dependencies, requiring the model to reconstruct the reasoning path from content rather than position.

As shown in Table 2 and Figure 5, the benefits of the reasoning vector persist robustly under all adversarial conditions. The enhanced model maintains a consistent 2-6% performance advantage over the baseline. Even when faced with noise and structural shuffling designed to break simple heuristics, the vector-enhanced model consistently outperforms the baseline. This provides strong evidence that the transferred capability is a fundamental improvement in systematic problem-solving, rather than a brittle, memorized pattern.

## 4.4 ABLATION STUDIES

To investigate the properties and direct impact of the reasoning vector, we conducted a series of systematic ablations on the 1.5B model using the GSM8K benchmark.

**Vector Removal Analysis.** In a critical test, we subtracted the reasoning vector from the baseline model ($\theta_{\text{degraded}} = \theta_{\text{base}} - \mathbf{v}_{\text{reason}}$). As shown in Table 3, this operation resulted in a catastrophic performance collapse. Accuracy on GSM8K plummeted to 33.4%, an 11.8% drop relative to the baseline. This strong, symmetric effectwhere addition improves performance and subtraction severely degrades itunderscores the vector's significant contribution to the model's reasoning abilities.

**Scaling Analysis.** We investigated the effect of the scaling factor $\alpha$ from Equation 2, testing values in $\{0.5, 1.0, 1.5, 2.0\}$. We found that $\alpha = 1.0$ achieved the optimal performance (50.0%), while $\alpha = 0.5$ yielded a smaller gain (47.2%), and values greater than 1.0 began to degrade performance (48.1% for $\alpha = 1.5$). This suggests the reasoning vector as extracted is naturally well-calibrated for direct transfer.

Table 2: **Robustness evaluation on Qwen2.5 1.5B.** The reasoning vector maintains its advantage even under challenging, custom-designed perturbations of the GSM8K dataset.

| Configuration | GSM8K | Hard Lite | Noise+Digit | Shuffle |
|---|---|---|---|---|
| Baseline | 45.7 | 38.2 | 41.0 | 40.6 |
| + Vector | 47.7 (+2.0) | 41.9 (+3.7) | 44.3 (+3.3) | 43.2 (+2.6) |
| + Vector + Think | **49.4** (+3.7) | **43.8** (+5.6) | **46.8** (+5.8) | **45.5** (+4.9) |

**Cross-Domain Transfer.** To assess generalization, we tested the transferability of reasoning vectors between domains. A vector derived from code-based RL training (on HumanEval) improved GSM8K

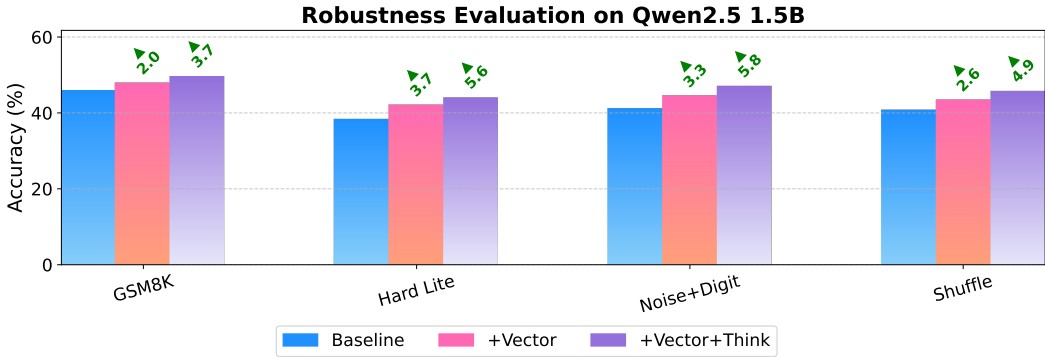

Figure 5: **Robustness of Qwen2.5 1.5B under four perturbation conditions.** Blue bars show baseline, orange is vector-enhanced, and green adds a reasoning prompt. The vector provides consistent gains across all conditions.

performance by 2.1%. Conversely, the math-derived vector from GSM8K improved HumanEval performance by 1.8%. While these cross-domain gains are smaller than in-domain gains, their existence indicates that the extracted vectors capture some domain-general components of reasoning. The ablation studies provide strong evidence for the vector's direct impact on reasoning, its well-calibrated nature, and its potential for modest generalization.

Table 3: **Ablation study on GSM8K using Qwen2.5 1.5B.** Subtracting the vector causes a severe performance drop (red), while adding it consistently improves results (green), highlighting the vector's strong impact.

| Model Configuration | Think Prefix | Accuracy (%) |
| --- | --- | --- |
| SFT Baseline | No | 45.1 |
| GRPO Donor | Yes | 46.8 |
| **+ Reasoning Vector** | No | 47.7 (+2.6) |
| **+ Vector + Think** | Yes | **50.0** (+4.9) |
| **- Vector (Subtracted)** | Yes | 33.4 (-11.8) |

## 5 LIMITATIONS

While our findings demonstrate the promise of reasoning vectors, it is crucial to acknowledge the limitations of this work.

**Architectural and Initialization Constraints.** The success of our method hinges on strict compatibility between the donor and target models. Effective transfer requires identical architectures, tokenizers, and a shared pre-training initialization family to ensure the parameter spaces are sufficiently aligned. Transferring reasoning vectors across different model families (e.g., from a Llama model to a Qwen model) is not guaranteed to work and remains an important open question.

**Dependence on Existing Donor Models.** Our approach is framed around *reusing* prior computational effort, not eliminating it. The method is contingent on the public availability of suitable SFT and RL-tuned donor models that meet the compatibility criteria. The significant cost of creating these donor models in the first place is externalized, and the method's applicability is therefore tied to the richness of the open-source model ecosystem.

## 6 Conclusion

This work establishes that reasoning ability can be extracted and transferred as a compact task vector between compatible models. By isolating parameter differences between SFT and GRPO checkpoints, we demonstrate that $\mathbf{v}_{\text{reason}} = \theta_{\text{GRPO}} - \theta_{\text{SFT}}$ captures transferable cognitive capabilities that generalize across diverse domains.

Our key finding is that the reasoning behaves as a modular and transferable component in parameter space. Consistent improvements in mathematical reasoning (GSM8K, +4.9%), code generation (HumanEval, +4.3%), and logical deduction (BigBenchHard, + 12.3%) indicate that the vector encodes domain-general problem solving strategies. Symmetric effects in ablation studies provide compelling evidence: Adding $\mathbf{v}_{\text{reason}}$ improves performance, while subtracting it causes severe degradation (-11.8% in GSM8K), demonstrating that reasoning capabilities exist as manipulable directions in parameter space.

Rather than requiring costly RL training for each target model, practitioners can now enhance reasoning through two tensor operations completing in seconds. This transforms reasoning enhancement from compute-intensive training to lightweight model editing, democratizing access by leveraging existing open-source checkpoints. The success of cross-domain transfer, where math-derived vectors improve code generation, reveals deeper connections between reasoning modalities.

Our work demonstrates that reasoning is a task vector that can be moved, combined, and reused, opening new avenues for efficient model enhancement in the open-source AI era.

## 7 Use of Language Models in Writing

We used large language models (LLMs) (ChatGPT, Gemini and Claude) to assist in drafting and refining the text of this paper. These tools supported clarity, coherence, and stylistic consistency throughout the writing process.

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

# A  APPENDIX

## A.1  BENCHMARK DETAILS

The BigBenchHard benchmark is a collection of challenging tasks designed to probe the limits of LLM reasoning. The tasks used in our evaluation span several domains.

Table 4: **Breakdown of tasks within the BigBenchHard benchmark.**

| Task | Domain | Problems | Complexity |
|---|---|---|---|
| logical_deduction_seven_objects | Logical deduction/constraint reasoning | 250 | High |
| temporal_sequences | Temporal/event-order reasoning | 250 | Medium |
| multistep_arithmetic_two | Multi-step word arithmetic | 250 | Medium |
| tracking_shuffled_objects_seven_objects | Object-tracking/working-memory | 250 | High |
| dyck_languages | Formal languages/bracket-matching | 250 | Medium |

# B  SAMPLE MODEL OUTPUTS SHOWING ABLATIONS

## B.1  PROBLEM STATEMENT

A fellowship program evaluates applicants through a sequence of stages:

1. **Stage 1 (Aptitude):** Out of 960 applicants, $\frac{5}{8}$ pass.

2. **Stage 2 (Technical):** Of those who passed Stage 1, $\frac{3}{5}$ pass while $\frac{2}{5}$ fail.

3. **Retake:** Of the Stage 2 failures, $\frac{1}{4}$ are invited to retake the test, and $\frac{2}{3}$ of them pass.

4. **Portfolio Review:** From the Stage 2 failures, a subset is invited to submit portfolios, of which half succeed.

5. **Final Interview:** $70\%$ of all applicants reaching this stage are awarded the fellowship.

*Question: How many applicants ultimately receive the fellowship?*

## B.2  MODEL A  $\theta_{\text{base}} - \mathbf{v}_{\text{reason}}$

**Stage 1 (Aptitude)**

$$\text{Applicants} = 960, \quad \text{Pass rate} = \tfrac{5}{8}$$
$$\Rightarrow \text{Pass} = 960 \times \tfrac{5}{8} = 600$$

**Stage 2 (Technical)**

$$\text{Pass rate} = \tfrac{3}{5}, \text{ Fail rate} = \tfrac{2}{5}$$
$$\text{Pass} = 600 \times \tfrac{3}{5} = 360, \quad \text{Fail} = 600 \times \tfrac{2}{5} = 240$$

**Retake (from Stage 2 failures)**

$$\text{Invited} = 240 \times \tfrac{1}{4} = 60$$
$$\text{Pass} = 60 \times \tfrac{2}{3} = 40$$

**Portfolio (Side Door)** *Incorrectly applied to Stage 1 passers*

$$\text{Invited} = 600 \times \tfrac{1}{9} = \tfrac{200}{3} \approx 66.67$$
$$\text{Pass} = \left(\tfrac{200}{3}\right) \times \tfrac{1}{2} \approx 33.33$$

**Final Interview**

$$\text{Final pass rate} = \tfrac{7}{10}$$
$$\text{Final passes} \approx 33.33 \times \tfrac{7}{10} \approx 23.33 \;\Rightarrow\; \mathbf{23}$$

## B.3 MODEL B $\theta_{\text{base}} + \mathbf{v}_{\text{reason}}$

**Stage 1 (Aptitude)**

$$\text{Applicants} = 960, \quad \text{Pass rate} = \tfrac{5}{8}$$
$$\Rightarrow \; \text{Pass} = 960 \times \tfrac{5}{8} = 600, \quad \text{Fail} = 960 - 600 = 360$$

**Stage 2 (Technical)**

$$\text{Pass} = 600 \times \tfrac{3}{5} = 360, \quad \text{Fail} = 600 - 360 = 240$$

**Retake (from Stage 2 failures)**

$$\text{Invited} = 240 \times \tfrac{1}{4} = 60$$
$$\text{Pass} = 60 \times \tfrac{2}{3} = 40$$

**Portfolio (from Stage 2 failures)**

$$\text{Invited} = 360 \times \tfrac{1}{9} = 40$$
$$\text{Pass} = 40 \times \tfrac{1}{2} = 20$$

**Final Interview**

$$\text{Candidates} = 360 \;(\text{Stage 2 pass}) + 40 \;(\text{Retake pass}) + 20 \;(\text{Portfolio pass}) = 420$$
$$\text{Final pass rate} = \tfrac{7}{10}$$
$$\Rightarrow \; \text{Final passes} = 420 \times \tfrac{7}{10} = \mathbf{294}$$

**At-a-glance comparison**

| Model | Computation Path | Final Passes |
|---|---|---|
| $\theta_{\text{base}} - \mathbf{v}_{\text{reason}}$ | Portfolio misapplied to Stage 1 passers | **23** |
| $\theta_{\text{base}} + \mathbf{v}_{\text{reason}}$ | Correct aggregation at Final Interview | **294** |

These examples qualitatively illustrate the effect of the reasoning vector. Removing it causes the model to lose the correct sequence of steps and misapply rules (e.g. applying the portfolio stage to Stage-1 passers), yielding an implausibly small final count. Adding it restores a coherent solution path: The model tracks subsets correctly, aggregates at the final stage, and recovers the expected result. The purpose of these examples is illustrative rather than evaluativethey are not used to compute accuracybut they show that the vector reliably strengthens multistep reasoning. We also observe a secondary benefit: outputs become more structured and consistent in formatting after vector injection.