# OpenReview forum: "Reasoning Vectors: Transferring Chain-of-Thought Capabilities via Task Arithmetic"
_ICLR.cc/2026/Conference — ICLR 2026 Conference Withdrawn Submission_

### Official Review · Reviewer_5e6S · 2025-10-22

**Soundness:** 2
**Presentation:** 2
**Contribution:** 3
**Rating:** 4
**Confidence:** 3

**Summary:**

This paper introduces reasoning vectors, derived by taking the parameter difference between a GRPO-trained model and its SFT counterpart, and applying this offset to other compatible models. The approach enables the transfer of reasoning capabilities without additional training. Experiments on GSM8K, HumanEval, SciQ, and BigBenchHard demonstrate consistent improvements, with some cases even surpassing the original GRPO model. The authors attribute these gains to linear mode connectivity between the SFT and GRPO models in parameter space.

**Strengths:**

1. Simple, computationally efficient idea requiring no retraining.
2. Consistent improvement across several reasoning benchmarks.
3. Potentially impactful if the method generalizes across architectures.

**Weaknesses:**

1. Missing donor SFT baseline: The paper does not report the performance of the donor SFT model that was used to compute the reasoning vector, making it hard to measure the actual gain from SFT to GRPO or to verify whether the reasoning vector truly encapsulates the reinforcement-learning-induced improvements.
2. No training details for SFT or GRPO: The descriptions of both supervised fine-tuning and GRPO optimization are overly abstract, lacking crucial information such as reward formulation, rollout strategy, learning rates, or number of training steps, which prevents reproducibility and weakens methodological credibility.
3. Figure 4 and Table 1 duplicate results: Figure 4 visually replicates the exact data already presented in Table 1 without introducing additional analysis or insights, suggesting that the space could have been used for more meaningful ablations or visual comparisons.
4. The paper assumes that the difference between independently trained GRPO and SFT models—both initialized from the same base—captures the reasoning gain introduced by reinforcement learning. However, this setting is not equivalent to the standard sequential SFT to GRPO pipeline used in practice, where GRPO fine-tunes on top of SFT. Consequently, the extracted vector may not correspond to the true incremental improvements achieved through reinforcement learning, raising concerns about the practical validity and interpretability of the proposed reasoning vector.

**Questions:**

1. Quantifying donor–target differences:
The reasoning vector is derived from a GSM8K-trained SFT model and applied to a separate instruction-tuned target (Qwen2.5-Instruct). Could the authors report or quantify the performance gap between these models to demonstrate that the transfer generalizes beyond closely aligned SFT configurations?
2. Missing donor SFT baseline: The donor SFT model’s results are not reported, leaving it unclear how much GRPO improves over SFT or whether the extracted vector truly captures reinforcement-driven reasoning gains. Could the authors include this baseline to better support the claimed effectiveness?
3. Transferability and attribution of gains: Have the authors tested whether the same reasoning vector remains effective when applied to base or differently fine-tuned models trained on other datasets? Such experiments would help determine whether the observed gains stem from the RL-derived vector itself or from synergy with the instruction-tuned target.

---

### Official Review · Reviewer_8JwL · 2025-10-27

**Soundness:** 2
**Presentation:** 3
**Contribution:** 1
**Rating:** 2
**Confidence:** 3

**Summary:**

This paper explores the task vector mechanism for reasoning enhancement.

**Strengths:**

- The paper demonstrates quality improvements using the proposed method.

**Weaknesses:**

- I can't shake the feeling that the paper has very limited novelty. We all know that task vectors work and can be applied to extrapolate beyond some weights (such as the reasoning model in this paper). The idea that reasoning capabilities can be embedded into a compact vector is also not novel

**Questions:**

N/A

---

### Official Review · Reviewer_wk2i · 2025-10-28

**Soundness:** 1
**Presentation:** 3
**Contribution:** 1
**Rating:** 2
**Confidence:** 4

**Summary:**

The authors show that there may be extracted a "reasoning" vector as the difference of weights of a reasoning and non-reasoning models. This vector may then be reused to power another compatible model, thus reducing the adaptation costs.

**Strengths:**

* The paper is well written.
* It addresses a timely topic: the inexpensive induction of reasoning capabilities and the interpretation of model capabilities.

**Weaknesses:**

* The experimental setup should be substantially improved in both breadth and rigor — see the “Questions.”
* The novelty beyond what is presented in [1] appears limited.

**Questions:**

* I am interested in whether the conclusions transfer to other models (e.g., LLaMA 3.1). Evaluation only on the Qwen models is insufficient to justify your claim (see [2]).
* GSM8K is an old and saturated benchmark. Please present results on datasets such as MATH500, AIME24/25, AMC23, Minerva-Math, and OlympiadBench (in that order of priority).
* What are the scores for “Baseline + Think”?
* In my experience, gains of 2.6 are within the noise range for math benchmarks. I would like to see a standard deviation reported for all results. I understand that you use greedy decoding, but please consider what you can do here.
* How exactly do you design the perturbations? For example, what is meant by “extended numerical ranges and more reasoning steps”? Please also provide the details for the other perturbations.
* Can you compare the “vector removal” with a random vector removal?

[1] Ilharco et al. “Editing Models with Task Arithmetic”
[2] Shao et al. “Spurious Rewards: Rethinking Training Signals in RLVR”

**Details Of Ethics Concerns:**

Figure 2 is copied from Figure 1 in [1] with minimal changes, and I don’t see a reference.

---

### Official Review · Reviewer_xtRU · 2025-10-29

**Soundness:** 2
**Presentation:** 2
**Contribution:** 1
**Rating:** 2
**Confidence:** 5

**Summary:**

The paper proposes Reasoning Vectors, a simple way to transfer reasoning ability between language models by subtracting weights of a supervised model from a reinforcement-learned one and adding the resulting vector to new models. This yields consistent gains on reasoning benchmarks with minimal computation.

**Strengths:**

* Simple, elegant method requiring only weight arithmetic
* Reproducible and leverages open-source models, enhancing accessibility

**Weaknesses:**

* Evaluations lack statistical significance and multiple runs, and the reported performance gains are very small likely within the margin of noise suggesting that the observed improvements may not represent a real enhancement but rather random variation.
* The method is highly impractical in real-world settings because it requires donor and target models to have identical architecture and tokenizer, a condition rarely met even among models from the same family
* The method adds a reasoning vector derived from a single task to a highly similar Qwen-Instruct model already trained on that task and many others. The resulting metric gains are minimal, suggesting that the improvement likely reflects task-specific overfitting rather than genuine general reasoning enhancement potentially improving one benchmark at the cost of degrading performance on others.
* The idea of weight interpolation and vector arithmetic in model parameter space is not novel [1]

[1] Rofin et al, Linear Interpolation In Parameter Space is Good Enough for Fine-Tuned Language Models

**Questions:**

See weaknesses

---

### Official Review · Reviewer_kUh4 · 2025-11-01

**Soundness:** 2
**Presentation:** 2
**Contribution:** 1
**Rating:** 2
**Confidence:** 4

**Summary:**

This paper determines the difference vector supervise-fine-tuned models and RL-fine tuned models. The difference vector is then used (by addition) to steer the behavior of the model.

**Strengths:**

The paper follows an interesting idea. The experimental results are promising. Overall, the paper is relatively easy to read and understand.

**Weaknesses:**

In my opinion, the rationale of the paper is inconsistent: the paper argues that reinforcement learning would be (too) costly. However, in order to determine the reasoning vector, reinforcement learning is required.

The experimental evaluation is overall too weak. It only uses two models of the same type/provider. This is by far too less to make any generalizable statements. It also considers just one set of fine-tuning data. In fact, fine tuning on just one small dataset is not that compute intensive. The more interesting scenario (to me) would be if a model could be further improved using a similar method *after* specific reasoning capacities have already been achieved.

The paper only considers a single snapshot for determining the vector. This direction of this vector, however, will change over the course of the training process. The paper should investigate if the vector direction actually converges during the training. By continued training, also the overall directions could rotate.

The improvements in the evaluation as only moderate for most benchmarks. Did you check for statistical significance and random variations in the responses`.

The generalization capabilities from one domain with others is a contentious topic; the paper should include a balanced discussion with different view points, specifically given the limited experimental design in this paper.

**Questions:**

One top of discussing the weaknesses mentioned before:

Page 4: When are parameter spaces "sufficiently aligned"? What about distilled models, e.g.?

I was also somewhat surprised that the transfer is just using the addition of the vector and not some parameter alpha multiplied with the vector as a summand. Why that? Could we increase the effect by adding it multiple times?

---

### Note · Authors · 2026-01-25

I have read and agree with the venue's withdrawal policy on behalf of myself and my co-authors.